# Full-Length Transcriptome Sequencing and Identification of *Hsf* Genes in *Cunninghamia lanceolata* (Lamb.) Hook

Yuan Ji [1,2,†], Hua Wu [1,†], Xueyan Zheng [3], Liming Zhu [1], Zeli Zhu [1], Ya Chen [1], Jisen Shi [1], Renhua Zheng [4,*] and Jinhui Chen [1,*]

1   Key Laboratory of Forest Genetics & Biotechnology of Ministry of Education, Co-Innovation Center for Sustainable Forestry in Southern China, Nanjing Forestry University, Nanjing 210037, China; jsjiyuan@163.com (Y.J.); whua55@126.com (H.W.); zhulm20160918@163.com (L.Z.); zhuzeli322@163.com (Z.Z.); chenya3424@163.com (Y.C.); jshi@njfu.edu.cn (J.S.)
2   Jiangsu Collaborative Innovation Center of Regional Modern Agriculture & Environmental Protection, Huaiyin Normal University, Huai'an 223300, China
3   National Germplasm Bank of Chinese Fir at Fujian Yangkou Forest Farm, Shunchang, Nanping 353211, China; zxy0553@163.com
4   Fujian Academy of Forestry, Fuzhou 350012, China
*   Correspondence: zrh08@126.com (R.Z.); chenjh@njfu.edu.cn (J.C.)
†   These authors contributed equally to this work.

**Abstract:** *Cunninghamia lanceolata* (Lamb.) Hook. (Chinese fir) is an important timber species that is widely cultivated in southern China. However, the shallow root system and weak drought resistance of Chinese fir are not enough to cope with high temperature and drought. In recent years, molecular biology has been used to modify plants to make them more resilient. Therefore, improving heat and drought resistance of Chinese fir by molecular biology technology is one of the best choices, whereas fewer genetic information resources for *C. lanceolata* limit more comprehensive molecular studies. In this study, single-molecule full-length transcriptome (SMRT) sequencing technology was used to obtain full-length transcriptome data on Chinese fir. A total of 21,331 transcripts were obtained via co-assembly, and 11,094 gene sets were obtained via further de-redundancy. In addition, gene function annotation and gene structure analysis were performed. We also used these data to identify nine heat shock transcription factors (Hsfs) in Chinese fir, and heat stress transcriptome and real-time quantitative polymerase chain reaction (PCR) analyses revealed expression changes in response to heat stress, indicating that these may play roles in heat resistance. These studies have enriched the genetic information resources of Chinese fir, which may be utilized for further species promotion, improvement, and application.

**Keywords:** Chinese fir; full-length transcriptome; heat shock factor; heat stress; SMRT

## 1. Introduction

*Cunninghamia lanceolata* (Lamb.) Hook. (Chinese fir), which belongs to the Taxodiaceae family, is an important wood species [1]. *C. lanceolata* is endemic and has a widely cultivated area in China [2]. Chinese fir features fast growth and is of high economic value; its wood is mainly used as building or paper raw material [3]. Relevant statistics show that the planted area of Chinese fir in China is up to 11 million hectares, accounting for 12.9% of China's plantation area [4]. Although fir has a good market prospect, there is still a risk of economic loss. Due to the characteristics of shallow roots and poor water retention capacity of Chinese fir, it may not have strong resistance to drought and high temperature. However, because of the frequency of extreme weather events, the risk is rising. High temperatures, lack of rainfall, and drought are devastating to the economic trees that are planted in large areas [5]. Allen et al. expect tree deaths caused by warming and drought to become more widespread [6]. High temperatures cause drying of trees' leaves and crack

trunks, impeding their supply of nutrients and water, and speeding up transpiration. In addition, families with stronger resistance tend to survive, so it is necessary to improve the resistance of trees themselves. Traditional breeding usually obtains good character families through excellent tree breeding [7]. Improving plants from a molecular perspective is one of the most studied methods to date.

Molecular cloning is a method used to obtain gene resources. For example, Wu et al. cloned the *PSK* gene of *C. lanceolata* and found that it promoted root growth and adventitious root formation [8]. In addition, high-throughput sequencing can provide abundant genetic information resources, but there are still few omics-related studies on Chinese fir. Lin assessed the genome size and basic characteristics of Chinese fir using a survey [9]. Ji et al. obtained a small quantity of genomic resources of Chinese fir by constructing a BAC library [10]. Zheng obtained some chloroplast-related genetic resources through chloroplast genome sequencing [11]. Limited omics studies hinder molecular studies on Chinese fir. Because of its development, high-throughput sequencing has become an important partner in molecular research [12].

For species with scarce omics data resources, transcriptome sequencing is an effective method for enriching genetic data resources and a tool for molecular research [13]. Transcriptome sequencing is mainly based on the application of next-generation sequencing (NGS) [14]. For example, Illumina sequencing read lengths are generally only about 300 bp. Therefore, short reads obtained via sequencing require a large amount of splicing before transcripts can be formed, and it is thus difficult to distinguish single-base differences.

Full-length transcriptome sequencing is based on PacBio single-molecule real-time (SMRT) sequencing technology [15]. Compared with NGS methods, such as Illumina, the read length of full-length transcriptome sequencing has significantly improved, up to 10 kb [16], so the sequence measured with full-length transcriptome sequencing does not need to be assembled and alternative splicing and new transcripts can be detected more accurately. This is of great significance for screening target genes and subsequent gene function research. At present, the full-length transcriptome has been applied in a variety of woody plants, such as *Cephalotaxus oliveri* [17], *Nitraria tangutorum* [18], and *Ginkgo biloba* [19], with relatively good results. Therefore, full-length transcriptome sequencing will be beneficial for further study of Chinese fir.

Here, we used SMRT sequencing to generate the full-length transcriptome of Chinese fir. This enabled us to obtain a large amount of transcription data, which provide valuable resources for further study of gene function and regulatory mechanisms of Chinese fir.

## 2. Materials and Methods

### 2.1. Plant Materials

The plant materials used in this experiment were three whole clonal tissue culture seedlings of Chinese fir '6421'. The '6421' original stock plant was selected from the Yangkou Forest Farm, Shunchang, Fujian Province, China, in 1964. Tissue culture seedlings were grown at 23 °C, in a 16 h light/8 h dark light cycle, and with 60% air humidity. These seedlings with the same growth conditions were selected and quickly placed into liquid nitrogen, and then stored at −80 °C for RNA extraction.

### 2.2. RNA Extraction, Library Construction, and SMRT Sequencing

RNA from three Chinese fir seedlings were extracted from Chinese fir using an RNA extraction kit (Vazyme, Nanjing, China). Subsequently, 1% agarose gel electrophoresis was used to assess the degree of RNA degradation and whether there was contamination. The purity of RNA was determined with NanoDrop2000 (Nanodrop, Waltham, MA, USA), and the integrity of RNA was evaluated using Agilent 2100 (Agilent, Santa Clara, CA, USA).

Then, the RNA of the three samples was mixed according to the same amount and used for library construction. Oligo(dT) was used as primers to enrich mRNAs containing polyA tails and to reverse transcribe them into cDNA. Then, the cDNA was screened to construct the full-length transcriptome library. Finally, after digestion by exonuclease, the

unconnected connector at both ends of cDNA was removed, primers were combined, and DNA polymerase was bound to form a complete SMRT Bell library. After qualified library inspection, PacBio Sequel platform (PN:100-092-800-03) was used for sequencing.

### 2.3. Data Processing

Sequence data were processed using the SMRTlink software(Version 5.0, Menlo Park, CA, USA). The circular consensus sequence (CCS) was generated from subread BAM files. CCS.BAM files were collected as the output, and were then classified into full-length and non-full-length reads using the pbclassify.py script, ignore polyA false, and minSeq Length 200. The non-full-length and full-length fasta files produced were then fed into the cluster step, which performs isoform-level clustering (ICE), followed by final Arrow polishing.

### 2.4. Gene Structure Analysis

Gene structure analysis was performed using the TAPIS pipeline. The GMAP output bam format file and gff format file were used for gene and transcript determination. Alternative splicing events and alternative polyadenylation events were then analyzed. Fusion transcripts were determined as transcripts mapping to two or more long-distance range genes and were validated based on at least two Illumina reads.

### 2.5. CDS, TF, and lncRNA Analyses

Plant transcription factors were predicted using iTAK software [20]. The CNCI (Coding-Non-Coding-Index) [21], CPC [22] (Coding Potential Calculator, Version 2.0), Pfam-scan (Protein family scan) [23], and PLEK (predictor of long non-coding RNAs and messenger RNAs based on an improved k-mer scheme) [24] were used to predict the coding potential of transcripts.

### 2.6. Functional Annotation

Gene function annotation was performed using the non-redundant nucleotide database (Nr), protein family (Pfam), Swiss-Prot protein (Swiss-Prot), Clusters of Orthologous Groups of proteins (COG), eukaryotic Ortholog Groups (KOG), Gene Ontology (GO), and Kyoto Encyclopedia of Genes and Genomes (KEGG) databases after redundancy removal using CD-HIT software (Version 4.6.2) [25].

### 2.7. Identification and Multi-Segment Alignments of Hsf Genes

The sequences assembled using full-length transcriptomes were used as a database, and the HMMER software (V3.10) was used to search the typical Hsf protein domain (PF00447). Hsf proteins of *Arabidopsis* were used as a reference to search for the full-length transcriptional protein library using the blastp program, where the screening e values of both were $1 \times 10^{-5}$; then, the intersection of the two results and redundancy were removed to obtain the candidate genes. Subsequently, SMART (http://smart.embl-heidelberg.de/, accessed on 1 July 2022) and CDD (https://www.ncbi.nlm.nih.gov/Structure/cdd/wrpsb.cgi, accessed on 1 July 2022) were used to manually confirm whether the candidate genes were *Hsf* genes.

### 2.8. Motif and Phylogenetic Analyses

Motif analysis of the identified Hsf proteins was performed using the MEME online tool (https://meme-suite.org/meme/, accessed on 5 July 2022) and motif visualization was performed using Tbtools [26]. For phylogenetic analysis, Clustal Omega [27] was used to align the Hsf proteins of rice, *Arabidopsis*, and *C. lanceolata*, and Trimal software (Version 1.2 ) [28] was used to cut out redundant gaps. Finally, Beast2.0 [29] software was used to construct phylogenetic trees, and Figtree (Version 1.43) [30] was used to polish the phylogenetic trees.

### 2.9. Hsf Expression Analysisusingtranscriptome Data

Based on the unpublished transcriptome of heat stress in our laboratory, the expected number of fragments per kilobase of transcript sequence per million base pairs sequenced (FPKM) expression value of the *Hsf* gene was searched for and obtained, and the relative expression level of the gene was visualized using Tbtools [26].

### 2.10. Heat Stress and qRT-PCR Analyses

To verify the role of *Hsfs* in heat stress, Chinese fir tissue culture seedlings with consistent growth were selected for a heat stress experiment at 39 °C. Leaves were collected for RNA extraction at 1, 4, 8, 12, and 16 h, and normal tissue culture seedlings were compared with the control. A reverse transcription kit was used to reverse transcribe the extracted RNA into cDNA for qRT analysis. The relative expression was calculated using the $2^{-\Delta\Delta CT}$ method [31].

## 3. Results

### 3.1. Sequencing Data Statistics and De-Redundancy

To obtain full-length transcriptome information on *C. lanceolata*, we used 3 tissue cultured seedlings of *C. lanceolata* 6421 with the same growth status as materials, and extracted the total RNA from the roots, stems, and leaves. Subsequently, Agilent 2100 was used to detect the RIN value (RNA integrity number) (Table S1). Quality RNAs were pooled, and RNA was used and reverse transcribed to construct a library for SMRT sequencing. In total, 20.62 Gb of data were obtained, including 6,747,129 reads, most of which were distributed within the range of 1000 to 5000 bp (Figure 1), with an average read length of 3057 bp and sequencing N50 of 3420 bp (Table 1).

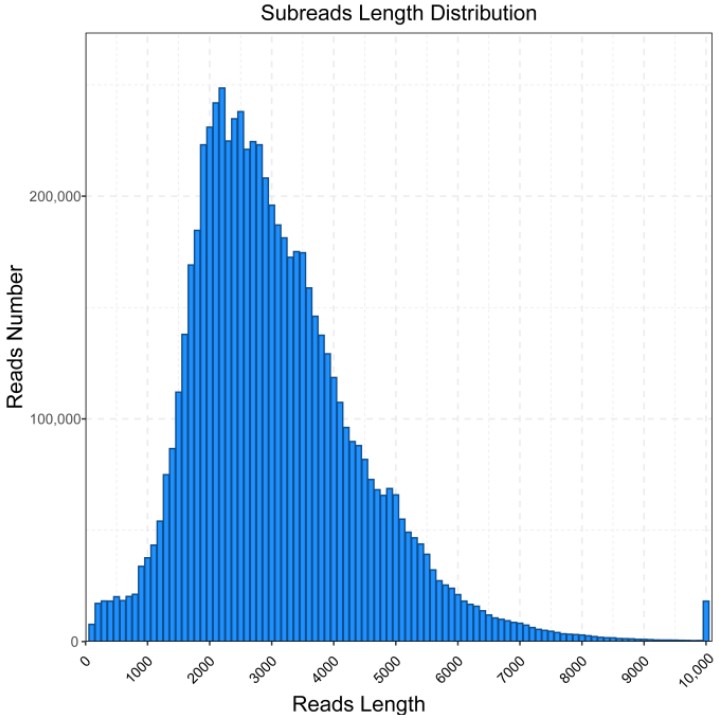

**Figure 1.** Distribution of subread lengths.

**Table 1.** Full-length transcriptome sequencing data.

| Sample | Subreads Base.G. | Subreads Number | Average Length | N50 |
| --- | --- | --- | --- | --- |
| Chinese fir | 20.62 | 6,747,129 | 3057 | 3420 |

### 3.2. Transcript Redundancy Analysis

Redundant, similar sequences tend to interfere with analysis, so we used CD-Hit software [32] to remove redundant sequences from the transcriptome. Starting from the longest sequence, the first cluster was formed, and then the sequence was processed in turn to complete the removal of redundant and similar sequences through clustering and comparison of protein or nucleic acid sequences. Table S2 shows the number of predicted genes, which was revised to 11,094 using CD-Hit software, a decrease of about 47% compared to before the redundancy. This indicated that there were many redundant sequences in the original splicing transcriptome, and thus it was necessary to remove the redundancy.

We also conducted statistical analysis of the full-length transcriptome data after redundancy was removed, and the results are shown in Table 2. The maximum length of the 11,094 genes was 10,329 bp, the minimum was 459 bp, and the average length was 3181 bp. The obtained de-redundant transcripts were sorted according to length, and the resulting N50 and N90 statistics were 3572 and 2000 bp, respectively.

**Table 2.** Length frequency distribution of transcripts before and after redundancy removal.

| Stage | <500 bp | 500 bp–1 kb | 1–2 kb | 2–3 kb | >3 kb | Total |
|-------|---------|-------------|--------|--------|-------|-------|
| Before | 6 | 107 | 3659 | 6718 | 10,841 | 21,331 |
| After | 3 | 78 | 2103 | 3461 | 5449 | 11,094 |

### 3.3. CDS, TF, and lncRNA Analyses

The coding sequence (CDS) is a sequence that encodes a protein product. Prediction of protein-coding regions is helpful for preliminary gene analysis and is also the basis for subsequent protein structure analysis. Therefore, ANGEL software [33] was used for CDS prediction analysis. A total of 11,157 CDSs were predicted, mainly between 500 and 3000 bp in length (Figure 2). Next, iTAK software was used to predict plant transcription factors, and the results showed that more than 800 transcription factors were detected. We plotted the number distribution of the top 30 transcription factors, among which C3H (58), PHD (43), and SNF2 (38) transcription factors were identified (Figure 3).

Then, CNCI (V2, default parameters), PLEK (V1.2, default parameters), and CPC2 (V0.1, default parameters) software and PfamScan (V1.6, default parameters) were used to predict the coding potential of sequencing data. CNCI, PLEK, CPC2, and Pfam predicted 349, 1288, 902, and 1982 lncRNAs, respectively. Subsequently, we conducted upset plot analysis of lncRNAs predicted by the four kinds of software and found that a total of 149 lncRNAs existed simultaneously (Figure 4).

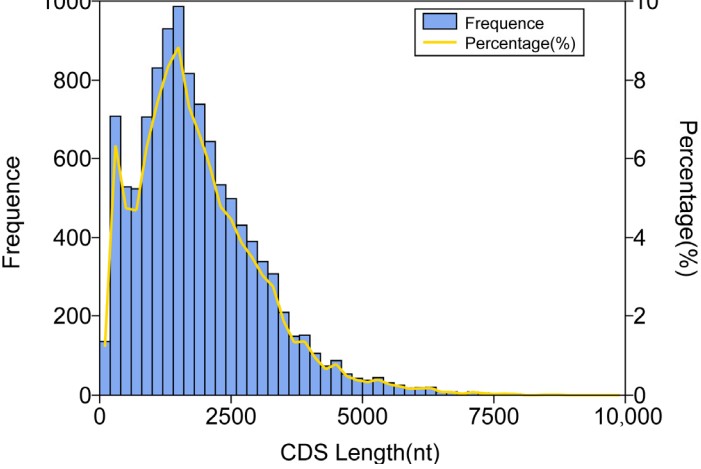

**Figure 2.** CDS length distribution.

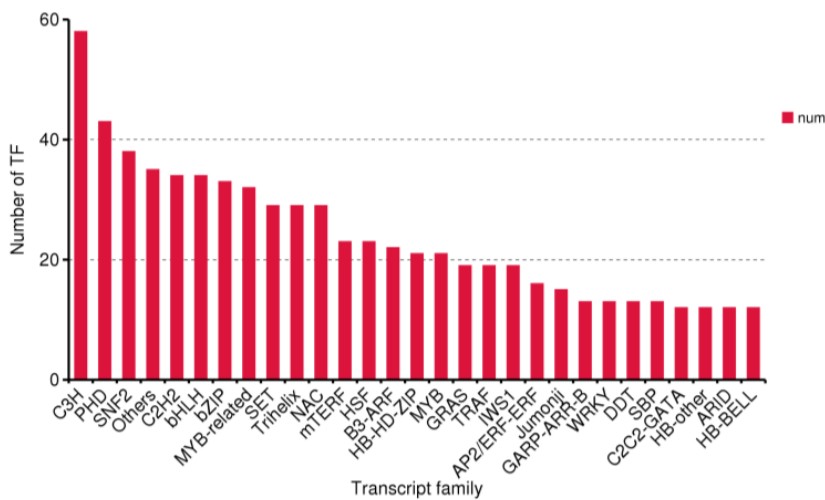

**Figure 3.** Transcription factor analysis.

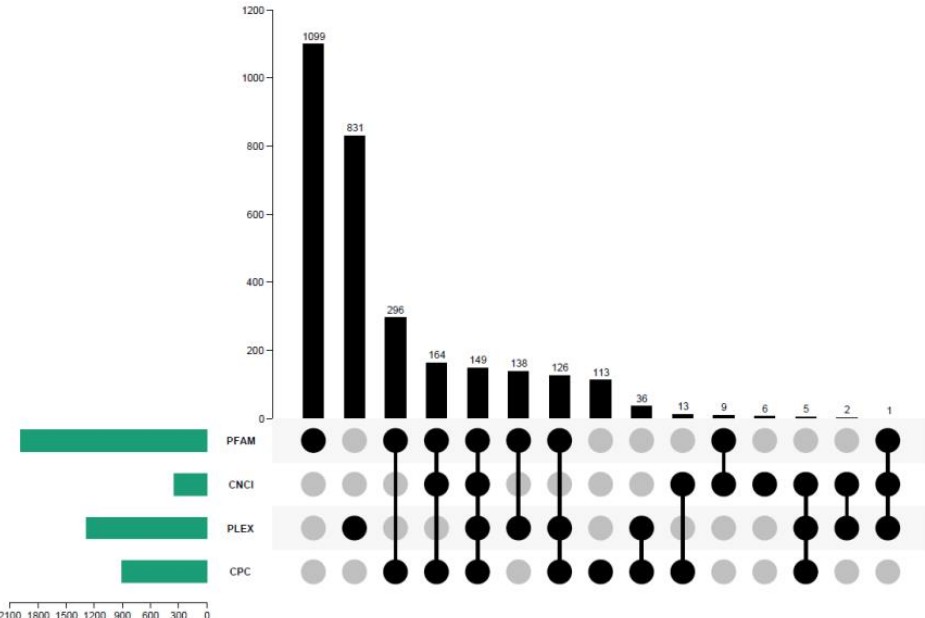

**Figure 4.** Upset plot of lncRNA prediction.

### 3.4. Functional Annotation of Genes

To obtain comprehensive gene function information, gene function annotation was performed on the sequences after redundancy removal using CD-Hit software. The Nr, Swiss-Prot, KEGG, KOG, GO, Nt, and Pfam databases were used. Approximately 11,094 transcripts were annotated, and the predicted transcripts from the Nr, Swiss-Prot, KEGG, KOG, GO, Nt, and Pfam databases accounted for 95.97%, 85.99%, 95.33%, 63.20%, 73.20%, 66.88%, and 73.20% of the total transcripts, respectively, with 10,723 genes annotated by at least one database (Figure S1, Table S3). In addition, using Venn diagrams we found that 4537 genes were simultaneously annotated in NR, NT, KOG, KEGG, GO, Swiss-Prot, and Pfam databases (Figure 5).

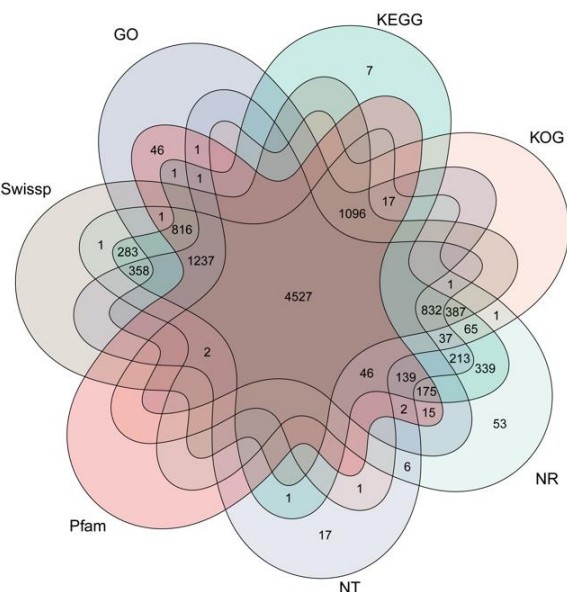

**Figure 5.** Venn diagram of functionally annotated genes.

The numbers in each large circle represent the number of transcripts of annotated genes in the database, and the part of the circles that overlaps represents the annotated genes shared among databases. According to the annotation results of the NR database, *Picea sitchensis* has the highest sequence matching degree with *C. lanceolata*, while *Amborella trichopoda* and *Nelumbo nucifera* have the highest similarity (Figure 6).

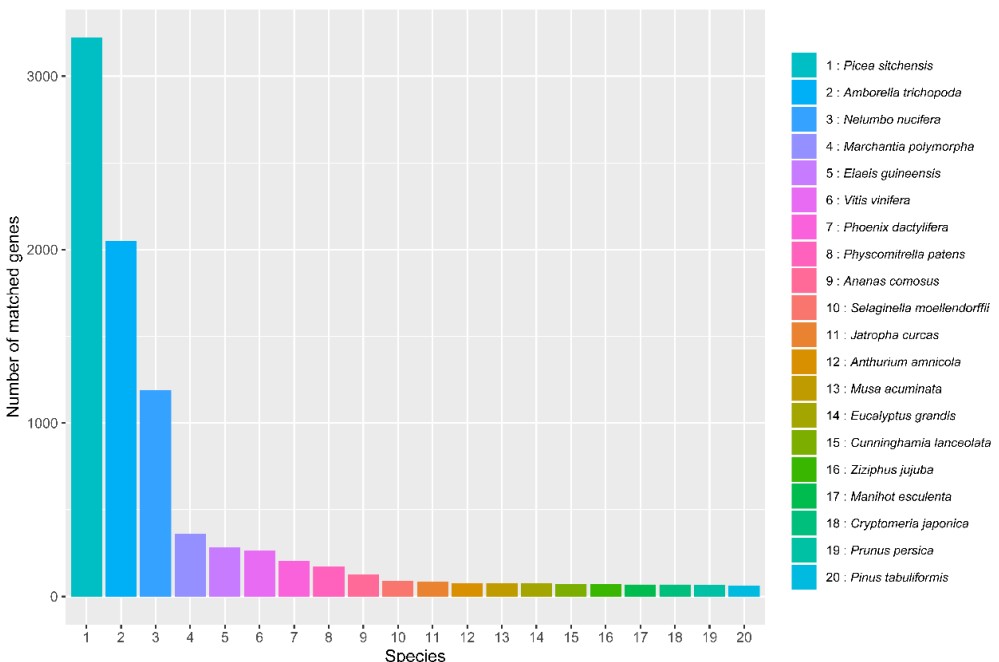

**Figure 6.** Genes annotated using NR databases.

The GO database was used to classify the annotated genes, and there were significant differences in three biological processes, including biological processes, cell components, and molecular functions. The functions of biological processes are mainly described in metabolic process and cellular process, and cell, cell part, organelle, and membrane in cellular component. Molecular functions, which focus on binding and catalytic activities, are shown in Figure 7.

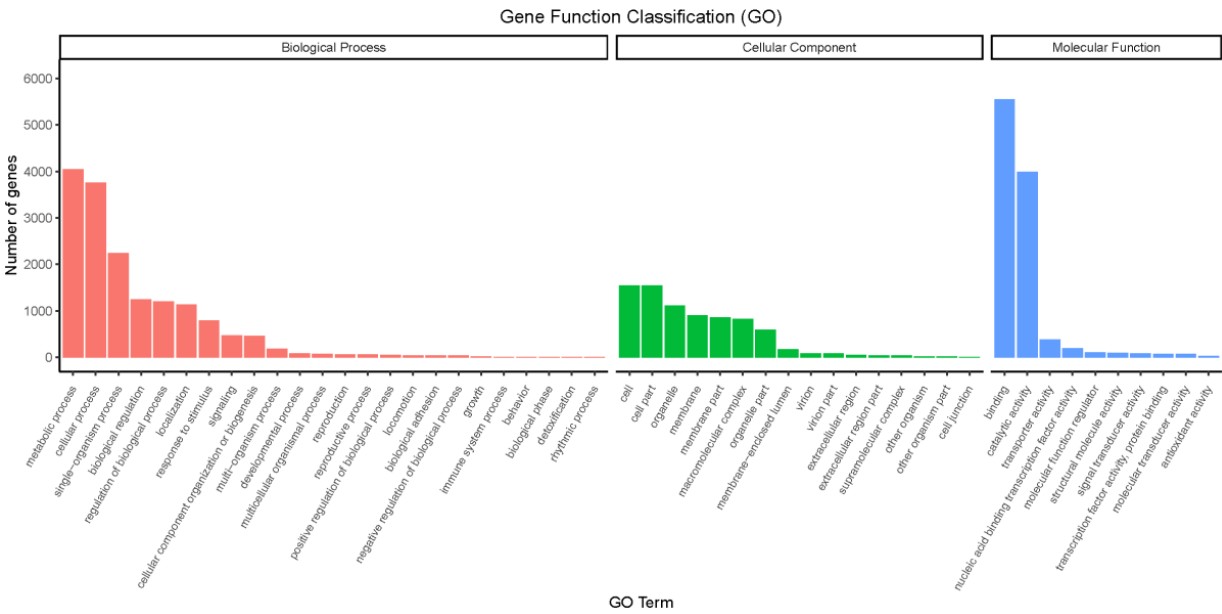

**Figure 7.** GO database annotation statistics.

Furthermore, we annotated the full-length transcriptome with the KOG database, and the 7011 annotated genes were associated with 26 processes such as RNA processing and modification, among which the general function prediction only (1449), posttranslational modification, protein turnover, chaperones (952), and T signal transduction mechanisms (681) were most abundant, while cell motility (10) and extracellular structures (12) were less abundant (Figure 8).

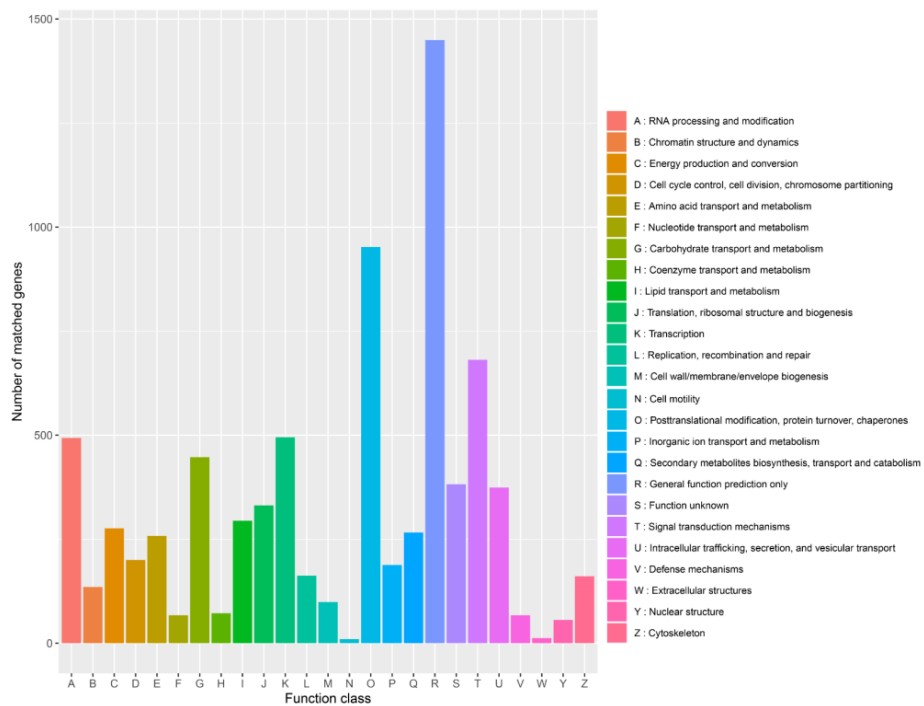

**Figure 8.** KOG database annotation statistics.

### 3.5. Identification of Hsf Genes Using the Full-Length Transcriptome Data

By combining HMMER and Blastp search results, we obtained 13 candidate Hsf genes. To prevent redundant sequences, referring to the screening method of Yao [34], we self-blasted the 13 candidates and removed sequences with a similarity >97%, and finally

identified 9 ClHsf genes. We named the 9 genes ClHsf-1 to ClHsf-9 according to their sequence of occurrence in the full-length transcriptome. Subsequently, we analyzed the basic characteristics of the nine identified ClHsf genes, including protein length, protein relative analysis quality, and isoelectric point (Table S4). Among these Hsf proteins, ClHsf-3, and ClHsf-7 were the smallest ClHsf genes identified, encoding a total of 317 amino acids, while the rest of the genes encoded from 319 to 524 amino acids. The relative molecular weight and isoelectric point analysis of the encoded proteins revealed that their relative molecular weights ranged from 35.16 to 58.78 kDa, and their isoelectric points ranged from 4.65 to 7.05 (Table S4). The Plant-mPLoc2.0 online tool (http://www.csbio.sjtu.edu.cn/bioinf/Cell-PLoc-2/, accesed on 7 July 2022) was used to predict their subcellular localization, which showed that these localized to the nucleus, suggesting that these are both typical transcription factors.

### 3.6. Conserved Domains and Phylogenetic Analysis

Motif sequences are usually closely related to the specific function of the protein family. To further analyze the function of the Hsf gene, we used MEME online tools to analyze the distribution of motif of ClHsf. As shown in Figure 9, Figure S2 motifs 1, 2, 4, and 7 existed in all Hsfs, indicating that these motifs were highly conserved, whereas motifs 3, 5, 6, 8, and 9 did not exist in ClHsf 6, 7, 9, indicating that these may have undergone functional differentiation during evolution. To further analyze the relationship among ClHsf genes, we constructed a phylogenetic tree (Figure 10) together with the Hsf genes of Arabidopsis and rice. The results showed that these Hsf genes were divided into six subgroups, of which ClHsf was distributed in four branches; clade I contained three ClHsfs, clade II contained two ClHsfs, clade V contained three ClHsfs, and clade III contained one ClHsf. In general, there are some differences in the motif distribution of ClHsf genes on different branches. This further indicates that these ClHsf genes may have undergone functional differentiation during evolution.

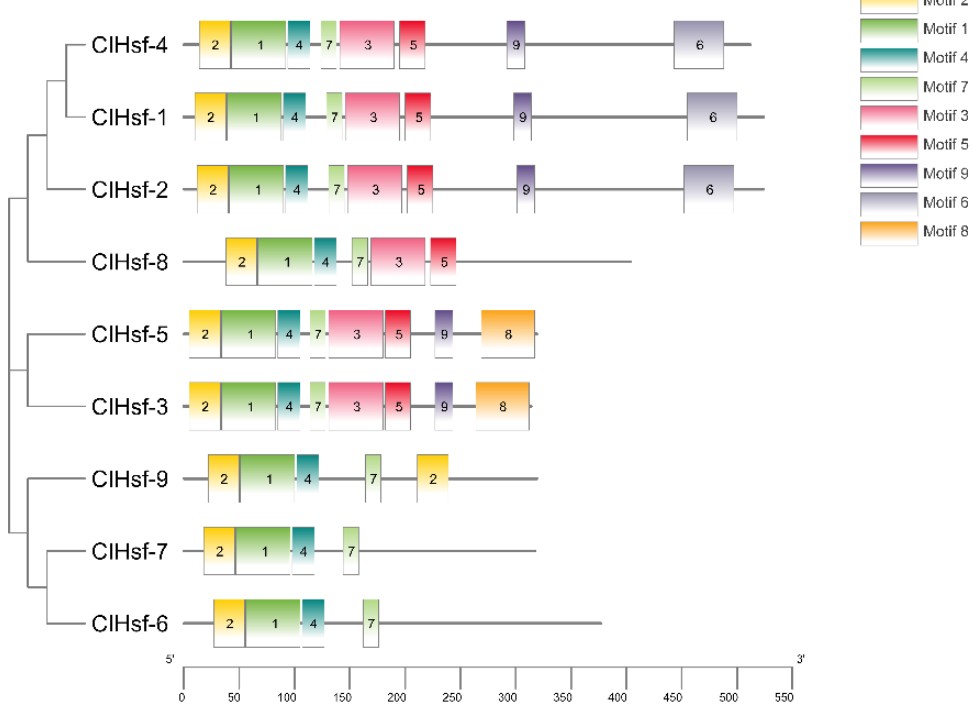

**Figure 9.** Conservative motifs of *Cunninghamia lanceolata Hsfs*.

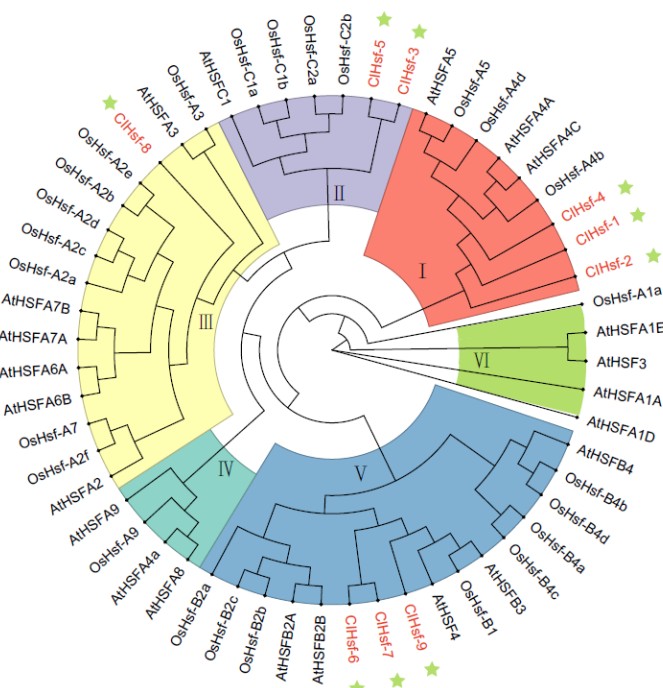

**Figure 10.** Phylogenetic analysis of *Hsfs* from *Arabidopsis*, rice, and *Cunninghamia lanceolata*. The green star represents the *Hsf* gene in *Cunninghamia lanceolata*.

*3.7. Expression of Hsfs in Transcriptomes under Heat Stress*

Previous studies have shown that Hsf plays a key role in plant tolerance to heat stress. To explore the role of ClHsfs in heat stress in Chinese fir, we used the unpublished transcriptome data of Chinese fir heat stress to explore the expression of its Hsf genes. The results showed that under 39 °C heat stress, the expression of the *ClHsf* gene in the leaves of Chinese fir seedlings showed an overall upward trend (Figure 11), and most Hsfs reached the maximum expression level at 1 h of stress, indicating that Hsfs begin to respond to heat stress signals at the early stage of Chinese fir under heat stress. Among these genes, *ClHsf-5* to *ClHsf-9* significantly increased by a factor ranging from a dozen to dozens of times compared with the control after 1 h of stress, and then the expression levels gradually decreased. Although the expression levels of the remaining Hsfs did not markedly increase, these were also significantly higher compared with the control. These results indicate that ClHsf gene expression is significantly increased when Chinese fir is subjected to heat stress, indicating that these Hsfs may play an important role in heat stress response.

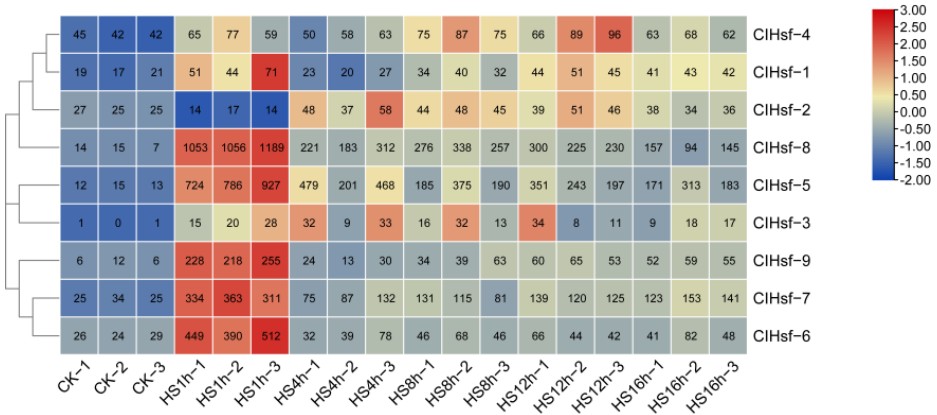

**Figure 11.** Heatmap analysis of *Hsfs* expression in the heat stress transcriptome.

### 3.8. Expression Patterns of Hsf Genes in Heat Stress and Different Tissues

To further determine the expression characteristics of *Hsf* in Chinese fir during heat stress, we selected four *Hsfs* with designed specific primers (Table S5) to detect their expression patterns of different branches under heat stress.

First, semi-quantitative PCR was performed to ensure that the primers of these HSP genes were specific (Figure S3), and then quantitative real-time PCR was used to detect their expression levels. The results showed that *ClHsf-1*, *ClHsf-5*, *ClHsf-8*, and *ClHsf-9* all responded significantly to heat stress and reached the peak expression at 1 h of heat stress (Figure 12), which was similar to the transcriptome expression pattern. This verified the reliability of our transcriptome data and indicated that *ClHsf-1*, *ClHsf-5*, *ClHsf-8*, and *ClHsf-9* may play a key role in heat stress responses.

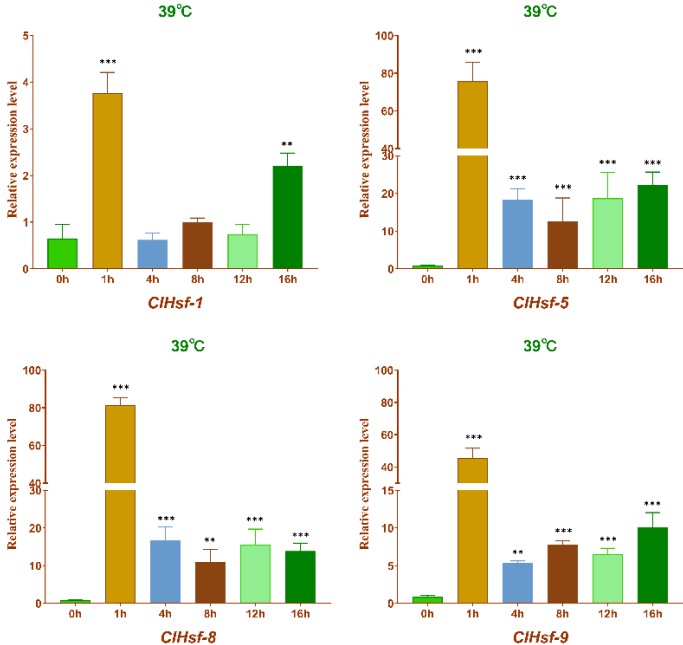

**Figure 12.** Expression analysis of *ClHsfs* under heat stress performed using q-RT. Analysis of variance was used for statistical analysis. ** $p < 0.01$, *** $p < 0.001$.

At the same time, we detected the expression differences of the four *ClHsps* in different tissues, and found that there was no significant difference in the expression of *ClHsf-1* in the roots and leaves, while the expression level of *ClHsf-5* and *ClHsf-9* in the stems was higher than that in the roots and leaves, and the expression level of *ClHsf-8* in the leaves was the highest (Figure S4). These results suggest that different Hsf members may play different roles in different tissues.

## 4. Discussion

Chinese fir is an economically significant wood species that is widely cultivated based on its characteristics of fast growth and good wood properties. However, this species is not resistant to severe cold, humidity, wind, or drought in its growth environment, thereby limiting its growth. With the rapid development of molecular technologies, molecular genetic improvement has become a powerful method for forest tree breeding.

However, limited genetic information resources restrict the molecular studies on Chinese fir. Traditional second-generation transcriptome sequencing has short transcript splicing and incomplete information. Therefore, it is necessary to obtain more accurate genetic information on Chinese fir. The PacBio Sequel-based SMRT sequencing has a maximum read length of 10 kb, which can effectively resolve issues relating to short transcript splicing and incomplete information associated with traditional second-generation sequenc-

ing such as Illumina. Complete transcripts can be obtained directly without interrupting splicing, thereby providing an important foundation for molecular research.

To date, full-length transcriptome information has been obtained for many species using the SMRT technology. For example, for sunflower, 10.43 Gb of clean data and 4,548,120 subreads were obtained using the full-length transcriptome [35], and for *Crocus sativus*, 11.3 Gb of data and 9,514,218 subreads were obtained [36]. In this study, a total of 6,747,129 subreads were obtained from 20.62 Gb of data using SMRT sequencing technology, and 21,331 transcripts were obtained via splicing. We clustered the corrected transcript sequences according to the 95% similarity between the sequences to remove redundancy and finally obtained 11,094 specific transcripts. We then performed structural analysis and functional annotation of these transcripts, which provided an important database for further molecular studies on Chinese fir.

Global warming has increased the frequency of extreme weather, such as extreme drought and extreme high temperatures, threatening the survival of plants [37,38]. The study of plant heat resistance has become an increasingly popular research direction. High temperature stress causes the plant chlorophyll to lose activity, reduces the rate of photosynthesis, and accelerates the evaporation of water inside the plant, resulting in water loss, drying, and ultimately death [39].Extremely hot weather is fatal to large-scale planted tree species such as Chinese fir without external water supply conditions in the wild. Therefore, it is necessary to study the adaptation mechanisms of plants under high temperature stress.

At present, many plant studies have shown that *Hsf* plays an important role in plant heat tolerance. For example, overexpression of *TaHsfA6f* can make transgenic wheat plants exhibit stronger heat tolerance [40], and *HsfA6b* regulates the response of Arabidopsis to heat stress through the ABA signaling pathway [41]. *Hsf* genes have also been identified in various species; for example, 29 *Hsf* genes were identified in Tartary buckwheat [42], of which 25 are present in maize [43] and 17 were identified in *Arachis* [44]. In our study, we identified nine *Hsfs* using the full-length transcriptome data of *C. lanceolata*, which is of significance in studying high temperature resistance in this species. We also analyzed the expression patterns of nine *Hsfs* under high temperature based on unpublished fir heat stress transcriptome data and real-time PCR, and found that they had a significant response in the initial period of heat stress (1 h) and a higher expression level during the heat stress process compared with the control. In general, this study provides a basis for further research on the molecular functions and regulatory mechanisms of *Hsfs*.

### 5. Conclusions

In this study, single-molecule full-length transcriptome (SMRT) sequencing technology was used to obtain full-length transcriptome data on Chinese fir. A total of 21,331 transcripts were obtained via co-assembly, and 11,094 gene sets were obtained via further de-redundancy. In addition, gene function annotation and gene structure analysis were performed. We also used these data to identify nine heat shock transcription factors (Hsfs) in Chinese fir, and heat stress transcriptome and real-time quantitative polymerase chain reaction (PCR) analyses revealed expression changes in response to heat stress, indicating that these may play roles in heat resistance. These studies have enriched the genetic information resources of Chinese fir.

**Supplementary Materials:** The following supporting information can be downloaded at: https://www.mdpi.com/article/10.3390/f14040684/s1, Figure S1: The number of genes obtained from different database annotations; Figure S2: motif structure map of ClHsps gene family; Figure S3: Semi-quantitative PCR of *ClHsf* gene family; Figure S4: Expression analysis of *ClHsfs* in different tissues. Table S1: RNA RIN values used for sequencing; Table S2: Sequence length distribution statistics table after de-redundancy; Table S3: Gene statistics annotated by different databases; Table S4: Physicochemical properties of ClHsf proteins; Table S5: primer for qRT-PCR of *ClHsf* gene family.

**Author Contributions:** Conceptualization and writing-original draft, Y.J.; data curation and visualization, H.W.; Formal analysis and validation, L.Z.; Writing—review and editing, X.Z.; Funding acquisition and investigation, Y.C. and Z.Z.; methodology, J.S.; project administration, J.C.; resources, R.Z. All authors have read and agreed to the published version of the manuscript.

**Funding:** This research was supported by the Seed Industry Innovation and Industrialization Engineering Project of Fujian Province (ZYCX-LY-202101), the Fujian Provincial Public-interest Scientific Institution Basal Research Fund (2020R1009003), the Nature Science Foundation of China (32071784), the Youth Foundation of the Natural Science Foundation of Jiangsu Province 632 (BK20210614) and Priority Academic Program Development of Jiangsu Higher Education Institutions (PAPD).

**Data Availability Statement:** The datasets supporting the conclusions and description of a complete protocol can be found within the manuscript and its additional files. The datasets used and/or analyzed during the current study are available from the corresponding author on reasonable request.

**Conflicts of Interest:** The authors declare no conflict of interest.

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
