# Peer review of "Full-Length Transcriptome Sequencing and Identification of Hsf Genes in Cunninghamia lanceolata (Lamb.) Hook"

_forests, doi:10.3390/f14040684_

Round 1

Reviewer 1 Report

my comments for this MS are as follws

1. author  must mention the genome size of the species and it ploidy level ?

2. line no 65. author shoul also mention the tissuese used in study properly, what was stage of the plant and how  many plant are considered to make ?

3.whether author has pooled the sample ? and how not mentioned ?

4.line no 66. modify the sentence for proper menaing like resistance to diseases and

5.in line o 67what was experimental condition and author should also mention about how many biological replicate per sample used in study along with RIN value sample wise. 

6.line 274. author should also enclosed the RT -PCR gel picture of the results as in supplementary files 

7.i am surprize still how author has done sampling without biological replicate 

8. Profiling of DEGs between differet tissues should be given in the study 

Author Response

my comments for this MS are as follws

  1. author must mention the genome size of the species and it ploidy level ?

Thank you for the comment. We have modified this part of the description according to your suggestions to make the article more smooth in line 58-65 as follows:” Molecular cloning is one of the methods to obtain gene resources. For example, Wu et al. cloned the PSK gene of Cunninghamia lanceolata and found that it promoted root growth and adventitious root formation[8]. In addition, high-throughput sequencing can provide abundant genetic information resources, but there are still few omics related studies on Chinese fir. Lin assessed the genome size and basic characteristics of Chinese fir through survey[9]. Ji et al. obtained scanty genomic resources of Chinese fir through by constructing BAC library[10]. Zheng has obtained some chloroplast-related genetic resources through chloroplast genome sequencing[11].

  1. line no 65. author shoul also mention the tissuese used in study properly, what was stage of the plant and how many plant are considered to make ?

Thank you for the comment. We used whole seedlings for this experiment, and we have added relevant descriptions in the revised manuscript in line 91 as follows: The plant materials used in this experiment were three whole clonal tissue culture seedlings of Chinese fir '6421'.

3.whether author has pooled the sample ? and how not mentioned ?

Thank you for the comment. We extracted RNA from three samples respectively, measured its concentration, and mixed it into a total RNA in the same proportion for the full-length transcriptional library construction. we have added relevant descriptions in the revised manuscript in line 103-104 as follows: Then the RNA of the three samples was mixed according to the same amount and used for library construction.

4.line no 66. modify the sentence for proper menaing like resistance to diseases and

Thank you for the comment. We feel that your suggestion is appropriate and we also think that this sentence is inappropriate in the material and method. We removed the relevant description from the revised manuscript.

5.in line 67what was experimental condition and author should also mention about how many biological replicate per sample used in study along with RIN value sample wise.

Thank you for the comment. We extracted RNA from 3 seedlings with the same growth for library construction in the method explained in Question 3. We also added the culture environment of Chinese fir seedlings to the revised manuscript in line 94 as follows: Tissue culture seedlings were grown with 23℃, 16-h light/8-h dark light cycle and 60% air humidity.

In addition, we added the RIN value of RNA used for sequencing in table S6. Meanwhile, lines 187-188 of the manuscript we were modifying were also added as follow: Subsequently, Agilent 2100 was used to detect the RIN value (RNA integrity num-ber)(Table S6).

6.line 274. author should also enclosed the RT -PCR gel picture of the results as in supplementary files

Thank you for the comment. According to your suggestion, we added ClHsf-1, ClHsp-5, ClHsf-8 and ClHsf-9 for semi-quantitative PCR results in different tissues and different heat stress treatment times, as shown in figure S4.

7.i am surprize still how author has done sampling without biological replicate

Thank you for the comment. We all used three biological replicates to construct the experiment. Due to our carelessness in writing, we did not describe the sample repeatability in the previous manuscript. We have added relevant descriptions of biological duplication to the revised manuscript.

  1. Profiling of DEGs between differet tissues should be given in the study

Thank you for the comment. According to your suggestion, we supplement the expression pattern of CIHsp in different organizations using qRT in the Figure S3. We have also added relevant descriptions in the revised manuscript in line 330-335 as follows: At the same time, we detected the expression differences of the four CIHsps in different tissues, and found that there was no significant difference in the expression of ClHsf-1 in the roots and leaves, while the expression level of ClHsf-5 and ClHsf-9 in the stems was higher than that in the roots and leaves, while the expression level of ClHsf-8 in the leaves was the highest (Fig S3). These results suggest that different hsf members may play different roles in different tissues.

Reviewer 2 Report

After reviewing the manuscript, I have to conclude that the authors have described the part results very accurately, but some parts of the manuscript (introduction and discussion) still need to be refined. In my opinion, the introduction should be informations that in the current version of the manuscript are in the discussion section; for example, general information about Cunninghamia lanceolata , the effect of global warming on plants

The informations given in l.313-315 is extremely important and shows the validity of the research - it should therefore be given in the introduction section. However, the sentence "in this study, we used the full-length transcriptome technology to determine the full-length transcript of Chinese fir superior clonal line 6421, analyze gene structure, and obtain functional gene annotation information(l.57-58) is a sentence that should be in the section „abstract”.

Keywords should not repeat words from the title.

In the abstract is a sentence "However, the relative scarcity of genetically related resources hinders further research on Chinese fir",- this is very important information., however, that is not addressed in the rest of the manuscript.

Now, in the introduction, the authors mainly describe the methods of molecular analysis of Cunninghamia lanceolata (Lamb.) Hook.(Chinese fir). However, I also have a comment relating to this description. The authors use the terms" Some scholars" (l.32), "Some research"(l.34) however, they cite only single publications in doing so.

References also still need work, for example, Latin names should be written in italics, pay attention to spaces.

Author Response

After reviewing the manuscript, I have to conclude that the authors have described the part results very accurately, but some parts of the manuscript (introduction and discussion) still need to be refined. In my opinion, the introduction should be informations that in the current version of the manuscript are in the discussion section; for example, general information about Cunninghamia lanceolata , the effect of global warming on plants

Thank you for the comment. In the revised version of the manuscript, we added information about the distribution and adaptation conditions of Chinese fir, and we also added content about the impact of global climate change on plants in line 33-45 as follows: Relevant statistics show that the planted area of Chinese fir in China is up to 11 million hectares, accounting for 12.9 of China's plantation area[4]. Although fir has a good market prospect, there is still a risk of economic loss. Due to the characteristics of shallow roots and poor water retention capacity of fir, it may not have strong resistance to drought and high temperature. But with the frequency of extreme weather events, the risk is rising. High temperatures, lack of rainfall, and drought are devastating to the economic trees that are planted in large areas[5].Allen et al. expect tree deaths caused by warming and drought to become more widespread[6]. High temperatures dry up trees' leaves and crack trunks, impeding their supply of nutrients and water, and speeding up transpiration. And families with stronger resistance tend to survive, so it is necessary to improve the resistance of trees themselves. Traditional breeding usually obtains good character families through excellent tree breeding[7]. Improving plants from a molecular perspective is one of the most studied methods to date.

The informations given in l.313-315 is extremely important and shows the validity of the research - it should therefore be given in the introduction section. However, the sentence "in this study, we used the full-length transcriptome technology to determine the full-length transcript of Chinese fir superior clonal line 6421, analyze gene structure, and obtain functional gene annotation information(l.57-58) is a sentence that should be in the section „abstract”.

Thank you for the comment. Extreme weather is an issue we need to pay attention to. We have added relevant introductions in the introduction according to your suggestions, and detailed descriptions have been added in the revised manuscript in line 39-44 as follows: High temperatures, lack of rainfall, and drought are devastating to the economic trees that are planted in large areas[5].Allen et al. expect tree deaths caused by warming and drought to become more widespread[6]. High temperatures dry up trees' leaves and crack trunks, impeding their supply of nutrients and water, and speeding up transpiration. And families with stronger resistance tend to survive, so it is necessary to improve the resistance of trees themselves.

There is some overlap between this part and the abstract part. According to your suggestion, we have rewritten the description of this part in lines 85-88 in the revised manuscript as follows: Here, we used SMRT sequencing to generate the full-length transcriptome of Chinese fir. This enables us to obtain a large amount of transcription data, which pro-vides valuable resources for further study of gene function and regulatory mechanisms of Chinese fir.

Keywords should not repeat words from the title.

Thank you for the comment. According to your suggestion, we removed and replaced the keywords that overlap with the title in line 23 as follows: Chinese fir; Full‐length transcriptome; Heat shock factor ; Heat stress; SMRT.

In the abstract is a sentence "However, the relative scarcity of genetically related resources hinders further research on Chinese fir",- this is very important information., however, that is not addressed in the rest of the manuscript.

Thank you for the comment. We have added relevant research references to the revised manuscript in line 61-66 as follows: but there are still few omics related studies on Chinese fir. Lin assessed the genome size and basic characteristics of Chinese fir through survey. Ji et al. obtained scanty genomic resources of Chinese fir through by constructing BAC library. Zheng has obtained some chloroplast-related genetic resources through chloroplast genome se-quencing. Limited omics studies hinder molecular studies on Chinese fir.

Now, in the introduction, the authors mainly describe the methods of molecular analysis of Cunninghamia lanceolata (Lamb.) Hook.(Chinese fir). However, I also have a comment relating to this description. The authors use the terms" Some scholars" (l.32), "Some research"(l.34) however, they cite only single publications in doing so.

Thank you for the comment. We revised the inappropriate description of the relevant research and rewrote the content of this section in the revised manuscript in line 58-65 as follows: Molecular cloning is one of the methods to obtain gene resources. For example, Wu et al. cloned the PSK gene of Cunninghamia lanceolata and found that it promoted root growth and adventitious root formation[8]. In addition, high-throughput se-quencing can provide abundant genetic information resources, but there are still few omics related studies on Chinese fir. Lin assessed the genome size and basic character-istics of Chinese fir through survey[9]. Ji et al. obtained scanty genomic resources of Chinese fir through by constructing BAC library[10]. Zheng has obtained some chloro-plast-related genetic resources through chloroplast genome sequencing[11]. .

References also still need work, for example, Latin names should be written in italics, pay attention to spaces.

Thank you for the comment. We checked all the species names in the references and italicized their Latin names. In addition, we added or removed some unnecessary spaces.

Reviewer 3 Report

I am glad that I had the opportunity to review the manuscript entitled "Full-Length Transcriptome Sequencing and Identification of Hsf Genes in Cunninghamia lanceolata (Lamb.) Hook." by Jinhui Chen and  Renhua Zheng.

The aim of the research presented in this manuscript was full-length transcriptome technology to determine the full-length transcript of Chinese fir superior clonal line 6421, analyze gene structure, and obtain functional gene annotation information

The manuscript is carefully prepared and quite well adapted to the template of the journal Forests. The Introduction is factual enough. The research is well conducted and quite well described in the Material and Methods chapter. The Discussion is also good enough. However, I have a few minor remarks that the Author should take into account:

1) The chapter Abstract lacks an outline of the research hypothesis, please complete it.

2) Please put keywords in alphabetical order.

3) There is no reference number 6 in the article, sources should be checked

4) As in line 138, "C.lanceolata" software should be changed so that there is a space between it.

5) As in line 162, a space should be left between the number and the abbreviation in "3.572bp" and similar expressions.

6) Species names in the reference list should be written in italics.

Author Response

1) The chapter Abstract lacks an outline of the research hypothesis, please complete it.

Thank you for the comment. According to your suggestion, we added the research hypothesis in the revised manuscript in line 13-17 as follows: However, the shallow root system and weak drought resistance of Chinese fir are not enough to cope with high temperature and drought. In recent years, molecular biology has been used to modify plants to make them more resilient. Therefore, improving heat and drought resistance of Chinese fir by molecular biology technology is one of the best choices. Whereas, fewer genetic information resources for Cunninghamia lanceolata limit more comprehensive molecular studies.

2) Please put keywords in alphabetical order.

Thank you for the comment. According to your suggestion, we have adjusted the order of keywords in alphabetical order in line 23 as follows: Chinese fir; Full‐length transcriptome; Heat shock factor ; Heat stress; SMRT.

3) There is no reference number 6 in the article, sources should be checked

Thank you for the comment. We have examined and supplemented the missing references 6.

4) As in line 138, "C.lanceolata" software should be changed so that there is a space between it.

Thank you for the comment. In the revised manuscript, we have replaced the representation of the Latin scientific name of Chinese fir to make it more standardized.

5) As in line 162, a space should be left between the number and the abbreviation in "3.572bp" and similar expressions.

Thank you for the comment. We have checked the full manuscript and added spaces in the revised manuscript between the number and the abbreviation.

6) Species names in the reference list should be written in italics.

Thank you for the comment. We checked all the species names in the references and italicized their Latin names.

Round 2

Reviewer 1 Report

Author has made many changes as suggested still i am expecting author to give the reply to my comments no 6.

 author should also enclosed the RT -PCR gel picture of the results as in supplementary files ? 

this is to confirm wheather these information can be utilized for future researchers ?

Author Response

We are very sorry that the supplementary diagram S4 is not easy to find because the supplementary diagram of the modified version is uniformly placed in one document(supplyment figure). In the supplementary figure S4, we used the cDNA of the roots, stems and leaves of Chinese fir and different time of heat stress as templates. From the figure, we could find that the overall expression trend of these 4 hsf genes was similar to the results of QRT, but there was no good visualization effect among some groups with low basic expression levels or not obvious expression differences. Therefore, this is also the reason why we chose fluorescence quantitative PCR with higher accuracy for further detection of HSF expression level.

yours

sincerely!
